# iPBS-Retrotransposon Markers in the Analysis of Genetic Diversity among Common Bean (*Phaseolus vulgaris* L.) Germplasm from Türkiye

**DOI:** 10.3390/genes13071147

**Published:** 2022-06-25

**Authors:** Kamil Haliloğlu, Aras Türkoğlu, Halil Ibrahim Öztürk, Güller Özkan, Erdal Elkoca, Peter Poczai

**Affiliations:** 1Department of Field Crops, Faculty of Agriculture, Ataturk University, Erzurum 25240, Türkiye; kamilh@atauni.edu.tr; 2Department of Biology, Faculty of Science, Cankiri Karatekin University, Çankırı 18200, Türkiye; 3Department of Field Crops, Faculty of Agriculture, Necmettin Erbakan University, Konya 42310, Türkiye; 4Health Services Vocational School, Binali Yıldırım University, Erzincan 24100, Türkiye; hiozturk@erzincan.edu.tr; 5Department of Biology, Faculty of Science, Ankara University, Ankara 06100, Türkiye; gulerozzkan@gmail.com; 6Department of Herbal and Animal Production, Ibrahim Çeçen University, Ağrı 04100, Türkiye; eelkoca@agri.edu.tr; 7Botany Unit, Finnish Museum of Natural History, University of Helsinki, P.O. Box 7, FI-00014 Helsinki, Finland; 8Institute of Advanced Studies Kőszeg (iASK), 9731 Kőszeg, Hungary

**Keywords:** bean, breeding, genetic diversity, population structure

## Abstract

Beans are legumes that play extremely important roles in human nutrition, serving as good sources of protein, vitamins, minerals, and antioxidants. In this study, we tried to elucidate the genetic diversity and population structure of 40 Turkish bean (*Phaseolus vulgaris* L.) local varieties and 5 commercial cultivars collected from 8 different locations in Erzurum-Ispir by using inter-primary binding site (iPBS) retrotransposon markers. For molecular characterization, the 26 most polymorphic iPBS primers were used; 52 bands per primer and 1350 bands in total were recorded. The mean polymorphism information content was 0.331. Various diversity indices, such as the mean effective allele number (0.706), mean Shannon’s information index (0.546), and gene diversity (0.361) revealed the presence of sufficient genetic diversity in the germplasm examined. Molecular analysis of variance (AMOVA) revealed that 67% of variation in bean germplasm was due to differences within populations. In addition, population structure analysis exposed all local and commercial bean varieties from five sub-populations. Expected heterozygosity values ranged between 0.1567 (the fourth sub-population) and 0.3210 (first sub-population), with an average value of 0.2103. In contrary, population differentiation measurement (Fst) was identified as 0.0062 for the first sub-population, 0.6372 for the fourth subpopulations. This is the first study to investigate the genetic diversity and population structure of bean germplasm in Erzurum-Ispir region using the iPBS-retrotransposon marker system. Overall, the current results showed that iPBS markers could be used consistently to elucidate the genetic diversity of local and commercial bean varieties and potentially be included in future studies examining diversity in a larger collection of local and commercial bean varieties from different regions.

## 1. Introduction

It has been reported that the rate of disappearance of plant species has increased in recent years and it is thought that the rate of genetic erosion of plant species will increase in the coming years [1]. To minimize genetic erosion in agriculture and to ensure sustainability in this field, many strategies have been developed for the protection of germplasm [2]. Germplasm refers to living tissues that are used in plant breeding studies and have a very important place for the conservation of plant genetic resources. One of the important tools in which plant germplasm is preserved is plant gene banks. These gene banks contain different plant germplasms such as seeds, pollen, in vitro. These gene banks are extremely important as they reflect the genetic diversity of both cultivated plants and their wild relatives [3]. Genetic variation information is crucial to GenBank management and breeding studies. This information assists in the creation of seed collections and facilitates the use of desired local varieties in breeding programs [4]. Knowledge of the genetic diversity between native species and improved varieties is crucial to supporting plant breeding programs so that breeders can take advantage of existing local varieties adapted to the climatic conditions of particular regions [5].

Bean (*Phaseolus vulgaris* L.) is one of the most valuable herbal products in the world due to its nutritional properties, benefits to human health and economic importance [1]. Beans are an important product that is widely grown and distributed in almost every region of the world [2]. Beans show wide variation phenotypically, biochemically, and genotypically, and are comprised of independent and differentiated gene pools, forming gene centers in Central America and the Andes Mountains [6]. The contributions of these two gene pools can generally be distinguished by seed size and certain other morphological characteristics. The seeds of the Mesoamerican local varieties are small or medium in size, while those of the Andean local varieties are larger [7]. The first bean cultivars corresponding to the small-grained Mesoamerican local varieties s were identified in Spain, Portugal, and South America in the early 16th century. Beans first came to Europe in the 16th-17th centuries [3]. It is reported that it reached Turkey from Europe in the 17th century [8]. Turkey is not the homeland of the bean, but several studies have indicated the existence of wide variation among local bean local varieties in Turkey [9]. The characterization of local varieties provides an opportunity to determine genetic diversity and to identify new variations that can be used in various breeding programs [10,11,12,13,14]. 

Genetic diversity studies have been carried out with bean varieties in many parts of Turkey. However, these studies are not yet enough. Such diversity studies can support breeding activities by both farmers and plant breeders. It is also crucial to the conservation and sustainable use of the plant genetic resources needed to meet future food-security demands [15]. 

Various morphological, chemical, biochemical, and molecular markers are widely used to characterize bean genetic diversity [16]. The development of molecular markers changed the fate of breeding studies and allowed these studies to accelerate. Molecular markers provide direct estimation of genetic variation at the DNA level, reducing the interference of environmental variation and being unaffected by the environment [17]. Molecular markers with different properties have been developed with studies by scientific communities. Various methods have used molecular markers, including amplified fragment length polymorphisms (AFLPs) [18], random amplified polymorphic DNA (RAPD) [19], sequence characterized amplified region (SCAR) [20], single nucleotide polymorphism (SNP) [21], inter simple-sequence repeat (ISSR) [4], simple-sequence repeats (SSR) [22], and expressed sequence tag (EST) [23], all to assess the genetic diversity and associations among several *Phaseolus* species. 

Moreover, among them, retrotransposons are genetic elements capable of forming major components of most eukaryotic genomes, constituting 50–90% of the plant genome. Retrotransposons are divided into two: long terminal repeat (LTR) and non-LTR retrotransposons. LTR-retrotransposons are more common in plants than the other group [24]. Due to limitations in both LTR and non-LTR retrotransposons, inter primer binding site (iPBS) retrotransposons have been developed as a universal marker used in the characterization of both animal and plant species [25]. iPBS markers are the dominant markers and have become a preferred marker in genetic diversity assessment in recent years due to their universality [26]. The universality of the iPBS-retrotransposon marker has been proven and molecular characterization and phylogenetic studies are available for these markers, also in beans [8,24,26]. In our previous studies [26] and in the studies of other researchers [8,9], it has been observed that retrotransposon markers are quite efficient for genetic diversity studies in terms of the total number of amplified and polymorphic bands. The local varieties evaluated so far represent only a small subset of the available resources. In addition, a comprehensive study has not yet been conducted to measure the genetic diversity of bean germplasm in Türkiye. Previous studies [7,8,9,20,26] allow the investigation of the genetic diversity of local bean varieties collected from a very narrow geographical region in Türkiye. There are no previous studies to reveal bean genetic diversity and population structure in Erzurum-Ispir district in the Northeastern Anatolia region of Türkiye using iPBS markers. Therefore, we here investigate the genetic diversity and population structure local bean varieties collected from the district of Ispir, using the iPBS marker system. It is necessary to identify, define, and use genetic resources for the continuity of breeding studies. We expect that our findings here will assist in the use, improvement, and preservation of local varieties that are well adapted to the changing environment. 

## 2. Materials and Methods

### 2.1. Plant Materials

In this study, 45 Turkish bean (*Phaseolus vulgaris* L.) local varieties were used as plant material. The names and gathering places of the regional varieties are presented in Table 1 and Figure 1. Bean local varieties were collected in cultivated fields in form eight different Ispir districts of Erzurum in the Northeastern Anatolia region of Türkiye. The plants were grown for tissue sampling in the greenhouse of Atatürk University, Department of Field Crops, Faculty of Agriculture.

### 2.2. DNA Isolation and Quantification

Young leaves of beans (*P. vulgaris* L.) approximately 15-day-old plants were ground in liquid nitrogen at the molecular biology and genetics laboratory of Ataturk University. The collective DNA of 45 individuals per participation was then prepared, using the DNA extraction method of Zeinalzadehtabrizi et al. [27], with modifications. The DNA quality was determined by electrophoresis, using agarose gel at 0.8% concentration. A NanoDrop ND-1000 UV/Vi’s spectrophotometer device (Thermo Fisher Scientific Company, Waltham, MA, USA) was used to determine the DNA concentrations. The final DNA concentration was selected for the iPBS analysis. The DNA samples for which the concentrations were determined were stored at –20 °C for PCR (polymerase chain reactions) after further dilution.

### 2.3. PCR and iPBS Marker Analyses

Genetic diversity analyses were performed with iPBS primers available from Sigma Aldrich (Castle Hill, NSW, Australia). In the present study, 26 iPBS primers developed by Kalendar et al. [28] were used (Table 2). PCR Amplification was performed in a thermos cycler (SensoQuest Labcycler) and were conducted in 10 µL reaction mixture comprising 25 ng template DNA, 0.5 U Taq polymerase, 0.25 mM dNTP, 1 µM (20 pmol) primer, 1X buffer; 2 mM MgCl2. The PCR thermal cycling profile is as follow; initial denaturation for 3 min at 95 °C, 38 cycles of 95 °C for 60 s, 50–60 °C for 60 s, 72 °C for 120 s and final extension at 72 °C for 10 min [29]. All PCR amplification products were resolved in agarose gel at 3% concentration at 200 V for 105 min. Finally, gels were visualized under UV light and photographed by digital camera (Model Nikon Coolpix500).

### 2.4. Data Scoring and Analysis

The PCR was performed in three replicates for each primer to verify the band pattern consistency. The DNA bands were scored, using TotalLab TL120 software (TotalLab Ltd., Gosforth, Newcastle upon Tyne, UK). For the iPBS amplification products, a band is scored “1” or absent “0” for each locus. Only clear, strong bands were scored, while faint, weak bands were ignored. The Numerical Taxonomy and Multivariate Analysis System for personal computer (NTSYSpc) V.2.0 programs based on the Dice similarity matrix [30] were used to determine the genetic similarities between the varieties. A UPGMA (Unweighted Pair-Group Method with Arithmetic mean) dendrogram was created with the NTSYSpc V.2.0 program. In addition, molecular variance (AMOVA) and PCoA (Principal Coordinate Analysis) analysis were performed using the Genalex 6.5 program [31]. A PIC (Polymorphism Information Content value) was used to assess the diversity of each iPBS marker [32]. The POPGEN v.1.32 program was used to determine the effective number of allele (ne), Nei genetic diversity (h), and Shannon’s information index (I) [33]. The Structure v.2.3.4 program was used to determine the genetic structures of the varieties [34,35]. Evanno’s ∆K [36] and Structure Harvester [37] methods were used to estimate the most expected K value. Using this method, Markov chain Monte Carlo (MCMC) posterior probabilities were estimated. The MCMC chains were run with a 10,000-iteration burn-in period, followed by 100,000 iterations using a model allowing for admixture and correlated allele frequencies. Principal coordinate analysis (PCoA) was performed with the GenALEx 6.5 program [38].

## 3. Results

### 3.1. Polymorphism Revealed by iPBS Primers

Sufficiently clear and scoreable bands were obtained from all primers included in the study. With these 26 primers, 1350 visible and scoreable bands were generated. The number of alleles in the primers varied between 23 (iPBS 2077 and 2383) and 80 (iPBS 2274) (Mean 37.14). When the analysis was performed with the iPBS markers, the PIC varied between 0.151 (iPBS 2298) and 0.495 (iPBS 2383) (Mean 0.331). Major allele frequency ranged from 0.528 (iPBS-2383) to 0.888 (iPBS-2298). The mean major allele frequency was 0.706 (Table 3).

### 3.2. Genetic Diversity

The number of effective alleles (ne), genetic diversity of Nei (h) and Shannon’s information index (I) value of the bean varieties is presented in Table 4. The greatest ne (1.720), h (0.419), and I (0.609) values were observed in variety G36. The lowest ne (1.470), h (0.320), and I (0.500) values were observed in variety G27. The mean ne, h, and I value were calculated as 1.566, 0.361, and 0.546, respectively.

### 3.3. Heterozygosity and Diversity of Varieties

The summary statistics for nine populations (na: Observed number of alleles, ne: effective number of alleles, I: shannon’s information index, He: expected heterozygosity, uHe: and unbiased expected heterozygosity are listed in Table 5. We determined that the He value ranged from 0.173 (Av) to 0.052 (Kt) (Mean 0.110), while the uHe value ranged from 0.104 (Kt) to 0.208 (Av) (Mean 0.149). The I value among the nine populations ranged from 0.072 (Kt) to 0.286 (Iov) (Mean 0.161). The Percentage of Polymorphic Loci (PPL) for bean was lowest at 10.38% and 13.21%. Among the nine populations of bean, the PPL value ranged from 10.38% (Mv) to 84.30% (Ic) (Mean 28.05%). The h values of the nine bean populations are presented in Table 6. Among the nine populations of bean from Ispir, the smallest h values observed were in Av/Uv (0.068), while the greatest were observed in Ic/Kv (0.232).

### 3.4. Principal Coordinate Analysis (PCoA) and Dendrogram Generated from 26 iPBS Markers

The unweighted pair-group method with arithmetic mean (UPGMA) dendrogram placed the 45-bean variety into three clusters. There were only 18 (40%), 14 (31.11%) and 13 (28.88%) varieties in the first to three clusteres, respectively (Figure 2). Cluster I contained 18 bean varieties including G36, G45, G44, G43, G42, G41, G40, G39, G38, G37, G35, G34, G33, G32, G30, G31, G29 and G28. Group II contained 14 bean varieties including G27, G26, G25, G24, G23, G22, G21, G20, G19, G18, G16, G15, G17 and G14. In addition, the third subcluster contained 13 bean varieties including G13, G12, G11, G10, G9, G8, G7, G4, G6, G5, G3, G2, and G1. Principal coordinate analysis (PCoA) spatially showed the relative h values between the varieties, revealing three distinct groups. All local varieties collected from Öztoprak Village of Ispir center and one local variety (G26) from Ağilere village are on the upper right, 2 varieties (G25, G27) from Ispir-Center, Maden village, Yeşilyurt and Ağıldere villages are on the lower left. The commercial varieties on the left of the axis and the varieties belonging to other locations are scattered in various parts of the diagram. The result showed the grouping pattern of the PCoA analysis corresponded with cluster analysis (Figure 3). The percentage of genetic diversity explained by each of the three main coordinates of the basic coordinate analysis was determined as 32.34, 6.35 and 5.23, respectively, and these first three components explained 43.92% of the diversity (Table 7). The group I contained G6, G8, G14, G7, G3, G9, G12, G13, G26, G1, G11, G4, G5, G2, G10, G15, G18, G27, G19, G16, G17, G21, G24, G20, G25, G22 and G23 where all of them consisted of Ispir-Öztoprak Village, Ağildere Village, Ispir-Center and Maden Village. The varieties within this group showed higher variation and were scattered over a larger area. The group II comprised of G41 (commercial variety) and G34 (Ulubel village). The third group was composed of all other accessions including G37, G42, G36, G45, G38, G44, G43, G40, G31, G33, G39, G32, G29, G35, G28 and G30. The results showed that G45, G44, G42, G37 and G 36 belong to groups II and III. AMOVA (Analysis of Molecular Variance) was used to detect the total variation and showed that the variation within populations was 67% and that between populations was 33% (Table 8).

### 3.5. Population Genetic Structure Analysis for iPBS Markers

To understand the population structure among the 45 bean varieties, we divided each entry into corresponding subgroups using the model-based approach in the STRUCTURE software. The ∆K value is used to calculate the optimum K value. The result of genetic structure analysis suggests that the greatest value of K was calculated as 5 (red [A], green [B], blue [C], yellow [D], pink [E]) (membership probability < 0.8) (Figure 4). At K = 5, group I included 1 variety containing G36 mixed with yellow and pink groups. Group II contained 7 varieties including G22, G23, G26, G25, G24, G20, G19. Group III included 12 varieties counting G8, G9, G11, G6, G3, G4, G7, G5, G10, G2 and G12. Group IV included 6 varieties counting G42, G41, G40, G38, and G43. Group V contains 4 varieties including G29, G30, G28 and G31. Furthermore, G21, G17, G16, G18, G27, G13, G14, G15, G1, G39, G44, G44, G37, G32, G33, G34 and G35 were placed in mixed groups (40.00%; membership probability < 0.8). The F-statistic (F_ST_) value was determined as 0.0002, 0.4371, 0.4061, 0.6372, and 0.5440 in the first to fifth subpopulations, respectively. Likewise, the expected heterozygosity values (He) were determined as 0.3210, 0.1858, 0.1947, 0.1567, and 0.1907 in the first to fifth subpopulations, respectively (Table 9 and Table 10).

## 4. Discussion

Determining the genetic diversity levels of the germplasm of a plant species is essential for the designing and structuring of plant breeding programs [39]. Molecular markers such as iPBS for determining the genetic diversity and associations of varieties and accessions play important roles in targeted parental selection independent of environmental influences. Along with a role of retrotransposons in the diversification of genetic material, retrotransposon activation is reported to be one of the key factors involved in host adaptation to environmental changes [40]. In our study, polymorphic iPBS markers enabled the identification of bean (*P. vulgaris* L.) species at the molecular level. This provided important information about the genetic associations between these varieties. The information produced by the iPBS marker system suggests that it can be used effectively for diversity studies and genetic analysis in bean varieties. Using this marker system, other researchers have successfully examined similar bean species in genetic diversity studies [6,8,24]. The genetic diversity observed in our study is higher than in similar studies performed on Turkish beans, using different molecular markers [7]. This result clearly indicates that iPBS retrotransposons are highly polymorphic markers. The PIC value is a crucial piece of information that scores the efficacy of polymorphic loci and indicates the discriminatory power of a primer [41]. In our study the PIC varied between 0.151 (iPBS 2298) and 0.495 (iPBS 2383) (Mean 0.331). In a similar study of beans in which iPBS markers were used, PIC values were reported between 0.19 and 0.42 (Mean 0.33) [26]. The results are different to those of [8], who found PIC values between 0.65 and 0.93 (Mean 0.80) in their study with iPBS retrotransposons in beans. The results of the researchers differed, probably due to the varieties being different, while other researchers used fewer markers. 

The mean number of effective alleles (ne), genetic diversity of Nei (h) and Shannon’s information index (I) value of the bean varieties were calculated as 1.566, 0.361, and 0.546, respectively. [42], in their study using iPBS markers in peas, reported I values between 0.24 and 0.58 (Mean 0.39). The mean PIC value (0.73) obtained in this study was higher than the studies performed on iPBS markers and guava (0.24) [43] and grape (0.44) [44]. According to the comparison, it can be said that the iPBS primers used in this study of beans are more suitable. The maximum number of effective alleles is always desirable as they indicate the presence of greater genetic variation. Moreover, Shannon’s index of knowledge is an important criterion for understanding variation, as it distinguishes genetic variation in a population combining abundance and uniformity. In a study to explore the genetic diversity and population structure of scarlet eggplant with iPBS markers, the average polymorphism information content was found to be 0.363. The mean effective number of alleles, mean Shannon’s information index and gene diversity values were reported as 1.298, 0.300 and 0.187, respectively [45]. The results differed, probably due to the plant species and the various locations studied. Knowledge of the genetic variation between populations of a plant species is crucial to breeding and conservation [46]. Population-specific traits within each bean strain or variety can also be used to optimize crossbreeding studies. 

The population structure identified in this study was consistent with distance-based clustering from principal coordinate analysis (PCoA). In our study, we showed that intraspecific crosses, especially those between the Ic/Kv (0.232), Yy/Ky (0.229), and Kt/Ic (0.222) populations, may produce stronger hybrids, due to their greater genetic distance. We also performed PCoA analysis to examine the genetic associations between bean varieties. In the first three axes, PCoA analysis explained 43.92% of the total variation. In PCoA analysis, cluster analysis data obtained from this matrix are generally considered reliable when the axes explain 25% or more of the total variation [47]. PCoA is a widely used method for assessing genetic diversity based on quantitative and qualitative traits that scales distance data to multidimensional planes to characterize diversity. However, the grouping based on population structure seems to be more accurate, as it could precisely differentiate the bean varieties. Molecular analysis of variance (AMOVA) revealed the presence of high variation within bean varieties, with the percentage of total variance being 67%. It has been stated that higher variations in varieties may be due to reasons such as selection, adaptation, gene flow, genetic drift, variation in ecotypes and pollination method [48].

The findings showed that the bean varieties were divided into five groups according to their genetic structures. Varieties accumulate several living mutations throughout the evolutionary process, which form the basis of genetic diversity. Moreover, recombination, random drift, natural selection, such forces shape the genetic makeup of populations. In the recent past, understanding population structure has become a feature of great interest, as it can be helpful in selecting various parents and mapping sign-trait relationships. As a tool, analysis of population structure can predict similarity levels between individuals, subpopulations, and contributions. When samples are plotted with different geographic origins, analysis of population structure shows the pattern of geographic distribution among populations [49]. In a similar study [9] reported that 67 bean varieties were divided into four subpopulations (K = 4). In a study by [50], SSR markers were used to determine the genetic diversity in 149 dry bean varieties, and the varieties were divided into three subpopulations (K = 3) according to genetic structure analysis. The markers used in the study are primarily effective in grouping the genotypes [17].

## 5. Conclusions

There are many tools for determining and revealing genetic diversity in plant breeding. However, in plant breeding programs, it is extremely important to know the genetic distance of the varieties that are not clearly defined are unknown in the germplasm. Although classical breeding studies have reached the desired rate in many plant species and varieties, molecular markers provide very important information in breeding programs in genotypes development studies. In addition, the determination of distance and proximity conditions between varieties by performing genetic analyses contributes to the creation of new populations and to obtaining high-yielding varieties with heterosis. Therefore, evaluation of the genetic diversity of local bean varieties is needed for the conservation and breeding of this genetic material. Molecular markers and genetic diversity studies provide the useful information that is so critically needed about population structure. More informative molecular markers, such as iPBS, are being increasingly used in the study of bean genetic diversity, and their power cannot be underestimated. In conclusion, we used the iPBS retrotransposon marker system to generate pre-breeding data that could potentially be applied to the identification of common bean (*P. vulgaris* L.) genetic re-sources, conservation, and selection of suitable parents to provide greater genetic diversity for use in breeding programs. We showed here that the iPBS marker system is a powerful and easy method for detecting variation among bean varieties. The current findings reveal the diversity in local bean varieties collected from Erzurum-Ispir and will provide a basis for subsequent bean breeding programs, as well as integrity in bean identification studies. According to the information obtained in the study, it was determined that the genetically most distant cultivars were the G1 and G36 local varieties. With future studies, it is thought that these varieties can be used in breeding and hybridization studies, taking into account their agro-morphological characteristics, their resistance to biotic and abiotic conditions. 

## Figures and Tables

**Figure 1 genes-13-01147-f001:**
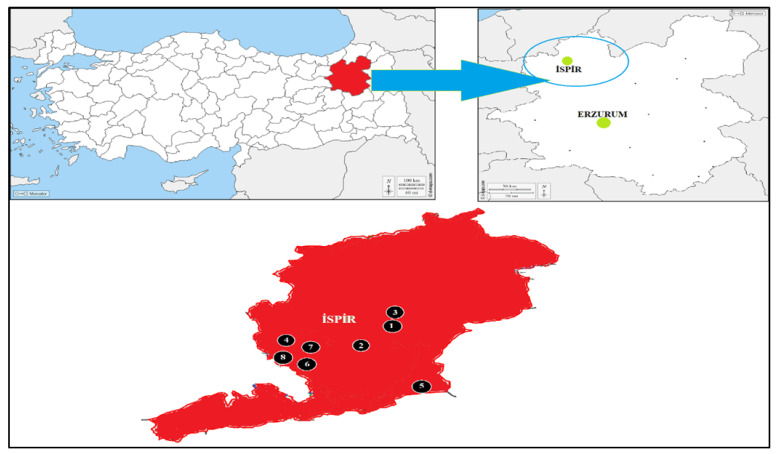
Locations where local bean varieties were collected (Table 1; 1: Öztoprak village, 2: Ispir Center, 3: Yeşilyurt, 4: Maden Village, 5: Elmalı District Ağıldere village, 6: Ulubel village, 7: Kirazlı village, 8: Köprübaşı town. Commercial cultivars are not shown on the map).

**Figure 2 genes-13-01147-f002:**
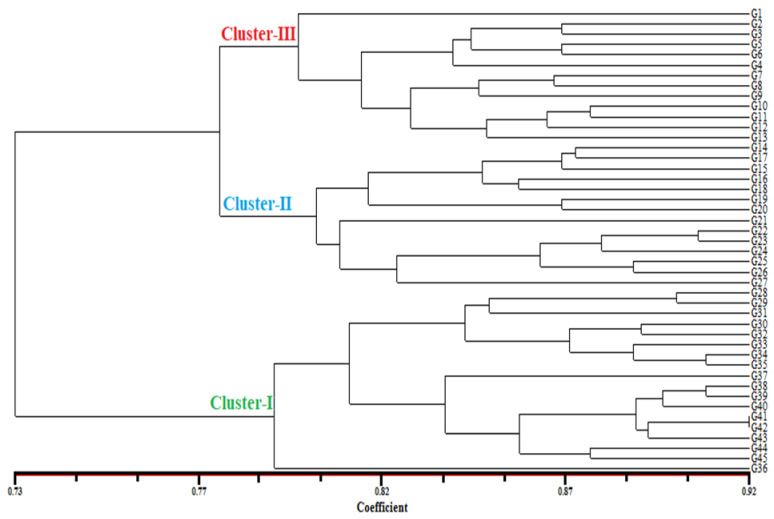
Dendrogram of 45-bean varieties generated with data from 26 inter primer binding site (iPBS) primers.

**Figure 3 genes-13-01147-f003:**
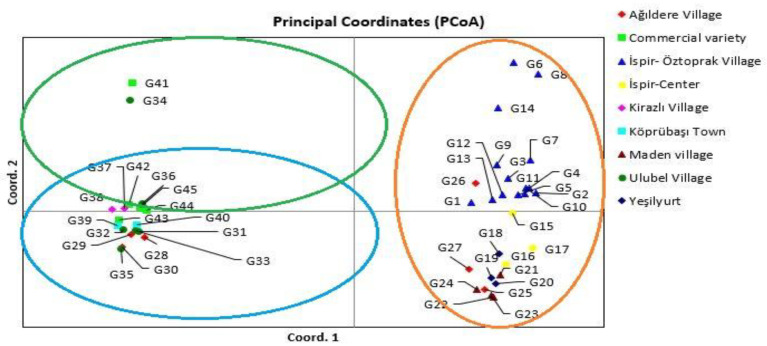
Principal coordinates analysis (PCoA) calculated from the pooled data of twenty-six inter-primer binding site (iPBS) primers in 45 bean varieties.

**Figure 4 genes-13-01147-f004:**
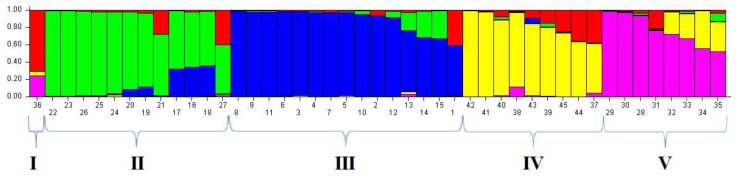
Genetic structure of varieties according to iPBS data (the bean varieties given in K = 5 (Figure 5) are presented in Table 9).

**Figure 5 genes-13-01147-f005:**
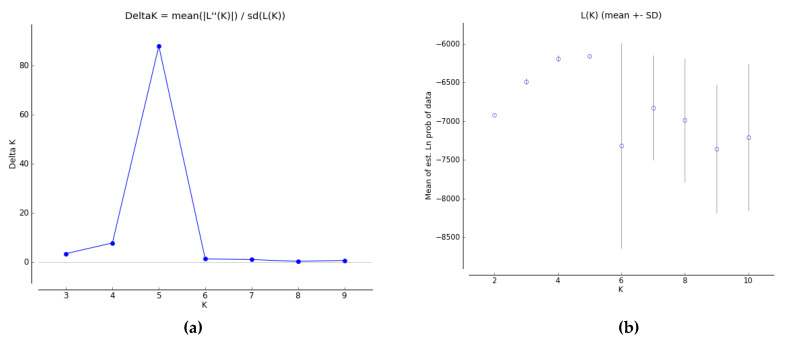
Line graphs from the mixture model of Ln P (D) and ∆K for bean populations (**a**); average value of the Ln P (D) statistics generated by the structure at each value of K (**b**); DK.

**Table 1 genes-13-01147-t001:** List of beans (*P. vulgaris* L.) local varieties and commercial cultivars collected from the Erzurum-Ispir district in Türkiye.

Variety	Collected Location	Latitude	Longitude	Altitude (m)
G1	Ispir- Öztoprak village	40.518	41.052	1431
G2	Ispir- Öztoprak village	40.518	41.052	1431
G3	Ispir- Öztoprak village	40.518	41.052	1431
G4	Ispir- Öztoprak village	40.518	41.052	1431
G5	Ispir- Öztoprak village	40.518	41.052	1431
G6	Ispir- Öztoprak village	40.518	41.052	1431
G7	Ispir- Öztoprak village	40.518	41.052	1431
G8	Ispir- Öztoprak village	40.518	41.052	1431
G9	Ispir- Öztoprak village	40.518	41.052	1431
G10	Ispir- Öztoprak village	40.518	41.052	1431
G11	Ispir- Öztoprak village	40.518	41.052	1431
G12	Ispir- Öztoprak village	40.518	41.052	1431
G13	Ispir- Öztoprak village	40.518	41.052	1431
G14	Ispir- Öztoprak village	40.518	41.052	1431
G15	Ispir-center	40.485	41.002	1264
G16	Ispir-center	40.468	40.983	1168
G17	Ispir-center	40.468	40.983	1168
G18	Yeşilyurt	40.518	41.069	1549
G19	Yeşilyurt	40.518	41.069	1549
G20	Yeşilyurt	40.518	41.069	1549
G21	Maden village	40.435	40.851	1226
G22	Maden village	40.435	40.851	1226
G23	Maden village	40.435	40.851	1226
G24	Maden village	40.435	40.851	1226
G25	Ağıldere village	40.401	40.834	1470
G26	Ağıldere village	40.401	40.834	1470
G27	Ağıldere village	40.401	40.834	1470
G28	Ağıldere village	40.401	40.834	1470
G29	Ağıldere village	40.401	40.834	1470
G30	Ağıldere village	40.401	40.834	1470
G31	Ulubel village	40.418	40.868	1424
G32	Ulubel village	40.418	40.868	1424
G33	Ulubel village	40.418	40.868	1424
G34	Ulubel village	40.418	40.868	1424
G35	Ulubel village	40.418	40.868	1424
G36	Ulubel village	40.418	40.868	1424
G37	Kirazlı village	40.436	40.887	1220
G38	Kirazlı village	40.436	40.887	1220
G39	Köprübaşı town	40.434	40.819	1286
G40	Köprübaşı town	40.434	40.819	1286
G41	Aras-98	Commercial cultivars		
G42	Elkoca-05			
G43	Göynük-98			
G44	Karacaşehir-90			
G45	Yakutiye-98			

**Table 2 genes-13-01147-t002:** List of 26 iPBS-retrotransposon primers with their sequence used to elucidate genetic diversity among 45 common bean varieties.

Marker	Primers Sequences (5′→3′)	Marker	Primers Sequences (5′→3′)
iPBS-2074	GCTCTGATACCA	iPBS-2377	ACGAAGGGACCA
iPBS-2077	CTCACGATGCCA	iPBS-2378	GGTCCTCATCCA
iPBS-2078	GCGGAGTCGCCA	iPBS-2380	CAACCTGATCCA
iPBS-2079	AGGTGGGCGCCA	iPBS-2381	GTCCATCTTCCA
iPBS-2080	CAGACGGCGCCA	iPBS-2383	GCATGGCCTCCA
iPBS-2095	GCTCGGATACCA	iPBS-2384	GTAATGGGTCCA
iPBS-2231	ACTTGGATGCTGATACCA	iPBS-2385	CCATTGGGTCCA
iPBS-2270	ACCTGGCGTGCCA	iPBS-2386	CTGATCAACCCA
iPBS-2271	GGCTCGGATGCCA	iPBS-2389	ACATCCTTCCCA
iPBS-2274	ATGGTGGGCGCCA	iPBS-2390	GCAACAACCCCA
iPBS-2276	ACCTCTGATACCA	iPBS-2391	ATCTGTCAGCCA
iPBS-2278	GCTCATGATACCA	iPBS-2392	TAGATGGTGCCA
iPBS-2298	AGAAGAGCTCTGATACCA	iPBS-2402	TCTAAGCTCTTGATACCA

**Table 3 genes-13-01147-t003:** Twenty-six iPBS primers used in the detection of polymorphism among 40 local varieties and 5 commercial cultivars of beans (*P. vulgaris* L.).

Marker	Number of Alleles	Major Allele Frequency	PIC *	Marker	Number of Alleles	Major Allele Frequency	PIC *
iPBS-2074	40	0.651	0.430	iPBS-2377	45	0.715	0.309
iPBS-2077	23	0.653	0.387	iPBS-2378	64	0.805	0.241
iPBS-2078	71	0.682	0.323	iPBS-2380	51	0.678	0.336
iPBS-2079	35	0.810	0.226	iPBS-2381	57	0.687	0.359
iPBS-2080	43	0.756	0.316	iPBS-2383	23	0.528	0.495
iPBS-2095	64	0.691	0.352	iPBS-2384	56	0.761	0.252
iPBS-2231	52	0.655	0.398	iPBS-2385	63	0.728	0.313
iPBS-2270	25	0.877	0.153	iPBS-2386	64	0.612	0.397
iPBS-2271	36	0.674	0.311	iPBS-2389	65	0.587	0.396
iPBS-2274	80	0.743	0.342	iPBS-2390	62	0.654	0.431
iPBS-2276	42	0.732	0.329	iPBS-2391	53	0.668	0.341
iPBS-2278	57	0.700	0.338	iPBS-2392	47	0.654	0.379
iPBS-2298	72	0.888	0.151	iPBS-2402	60	0.776	0.292
Mean					52	0.706	0.331

* PIC: Polymorphism Information Content.

**Table 4 genes-13-01147-t004:** Summary statistics for mean values for beans (*P. vulgaris* L.) varieties assessed with 26 iBPS primers.

Variety	ne *	h **	I *	Variety	ne *	h **	I *
G1	1.491	0.329	0.511	G24	1.530	0.347	0.531
G2	1.538	0.350	0.534	G25	1.586	0.369	0.556
G3	1.540	0.351	0.535	G26	1.550	0.355	0.540
G4	1.601	0.376	0.563	G27	1.470	0.320	0.500
G5	1.521	0.343	0.526	G28	1.658	0.397	0.586
G6	1.568	0.362	0.548	G29	1.696	0.410	0.601
G7	1.609	0.379	0.566	G30	1.642	0.391	0.580
G8	1.604	0.377	0.564	G31	1.688	0.408	0.598
G9	1.593	0.372	0.560	G32	1.588	0.370	0.557
G10	1.591	0.372	0.559	G33	1.586	0.369	0.556
G11	1.576	0.365	0.552	G34	1.524	0.344	0.528
G12	1.589	0.371	0.558	G35	1.476	0.322	0.503
G13	1.549	0.354	0.539	G36	1.720	0.419	0.609
G14	1.568	0.362	0.548	G37	1.648	0.393	0.582
G15	1.562	0.360	0.546	G38	1.520	0.342	0.526
G16	1.538	0.350	0.535	G39	1.567	0.362	0.548
G17	1.538	0.350	0.534	G40	1.528	0.345	0.529
G18	1.570	0.363	0.549	G41	1.564	0.361	0.546
G19	1.470	0.320	0.500	G42	1.562	0.360	0.546
G20	1.526	0.345	0.529	G43	1.586	0.370	0.556
G21	1.540	0.351	0.535	G44	1.556	0.358	0.543
G22	1.514	0.340	0.523	G45	1.505	0.335	0.518
G23	1.521	0.342	0.526	Mean	1.566	0.361	0.546

* ne: Number of effective alleles; ** h: genetic diversity of Nei; * I: Shannon’s information index.

**Table 5 genes-13-01147-t005:** Summary statistics for 45 bean (*P. vulgaris* L.) varieties assessed with 26 iPBS primers.

Population	n	na	ne	I	He	uHe	PPL (%)
Av	6	0.908	1.305	0.253	0.173	0.208	43.40
Iov	14	1.098	1.270	0.254	0.165	0.178	24.72
Ic	3	0.519	1.166	0.132	0.092	0.138	53.58
Kv	2	0.389	1.132	0.092	0.066	0.132	20.75
Kt	2	0.336	1.104	0.072	0.052	0.104	13.21
Mv	4	0.613	1.182	0.158	0.107	0.143	10.38
Uv	6	0.781	1.218	0.195	0.130	0.156	26.98
Yy	3	0.560	1.190	0.151	0.106	0.158	35.66
Com	5	0.574	1.165	0.142	0.096	0.120	23.77
Mean		0.642	1.192	0.161	0.110	0.149	28.05

n: number of sample size, na: number of distinct alleles, ne: effective number of alleles, I: Shannon’s information index, He: expected heterozygosity, uHe: unbiased expected heterozygosity, PPL: percentage of polymorphic loci; Av: Ağıldere village, Iov: Ispir-Öztoprak village, Ic: Ispir-center, Kv: Kirazlı village, Kt: Köprübaşı town, Mv: Maden village, Uv: Ulubel village, Yy: Yeşilyurt, Com: Commercial variety.

**Table 6 genes-13-01147-t006:** Pairwise population matrix of Nei genetic distance for nine groups of bean (*P. vulgaris* L.) varieties.

	Av	Com	Iov	Ic	Kv	Kt	Mv	Uv	Yy
**Av**	0.000								
**Com**	0.125	0.000							
**Iov**	0.124	0.179	0.000						
**Ic**	0.137	0.215	0.081	0.000					
**Kv**	0.128	0.072	0.209	0.232	0.000				
**Kt**	0.129	0.071	0.207	0.222	0.071	0.000			
**Mv**	0.099	0.202	0.114	0.109	0.215	0.211	0.000		
**Uv**	0.068	0.085	0.177	0.202	0.081	0.108	0.184	0.000	
**Yy**	0.119	0.207	0.104	0.086	0.229	0.212	0.087	0.197	0.000

Av: Ağıldere village, Com: Commercial variety, Iov: Ispir-Öztoprak village, Ic: Ispir-center, Kv: Kirazlı village, Kt: Köprübaşı town, Mv: Maden village, Uv: Ulubel village, Yy: Yeşilyurt.

**Table 7 genes-13-01147-t007:** PCoA analysis of bean varieties.

Axis	1	2	3
%	32.34	6.35	5.23
Cum %	32.34	38.69	43.92

**Table 8 genes-13-01147-t008:** AMOVA of bean varieties, using inter primer binding site (iPBS) marker.

Scheme	Degree of Freedom (DF)	Sum of Squares (SS)	Variance Component	% Of Total Variance	*p*-Value
Among Population	8	1150.70	21.439	33%	0.332
Within Population	36	1554.89	43.192	67%	0.001
Total	44	2705.60	64.631	100%	

**Table 9 genes-13-01147-t009:** Membership coefficients of five subpopulations of bean varieties.

Subpopulation							Subpopulation				
Varieties	I	II	III	IV	V	Varieties	I	II	III	IV	V
G1	0.401	0.005	0.579	0.004	0.012	G24	0.017	0.946	0.009	0.025	0.003
G2	0.059	0.005	0.923	0.008	0.006	G25	0.021	0.960	0.002	0.004	0.014
G3	0.009	0.002	0.972	0.013	0.004	G26	0.012	0.968	0.004	0.005	0.011
G4	0.014	0.012	0.970	0.003	0.001	G27	0.399	0.560	0.010	0.011	0.019
G5	0.011	0.011	0.961	0.011	0.006	G28	0.033	0.018	0.004	0.007	0.938
G6	0.008	0.003	0.975	0.011	0.003	G29	0.004	0.002	0.003	0.002	0.989
G7	0.024	0.002	0.969	0.002	0.002	G30	0.009	0.004	0.005	0.004	0.979
G8	0.002	0.003	0.993	0.001	0.001	G31	0.214	0.003	0.005	0.010	0.767
G9	0.007	0.009	0.980	0.003	0.002	G32	0.010	0.004	0.003	0.257	0.727
G10	0.007	0.041	0.946	0.003	0.003	G33	0.011	0.024	0.006	0.286	0.674
G11	0.005	0.010	0.979	0.003	0.003	G34	0.002	0.002	0.002	0.432	0.561
G12	0.014	0.070	0.909	0.004	0.003	G35	0.030	0.095	0.006	0.342	0.528
G13	0.025	0.205	0.709	0.031	0.030	G36	0.702	0.002	0.003	0.046	0.246
G14	0.013	0.298	0.682	0.003	0.004	G37	0.378	0.004	0.002	0.572	0.043
G15	0.007	0.320	0.665	0.004	0.004	G38	0.009	0.006	0.009	0.857	0.118
G16	0.017	0.640	0.336	0.005	0.002	G39	0.150	0.041	0.005	0.792	0.012
G17	0.003	0.670	0.323	0.002	0.002	G40	0.078	0.028	0.007	0.870	0.017
G18	0.014	0.625	0.344	0.007	0.009	G41	0.009	0.004	0.003	0.984	0.002
G19	0.031	0.849	0.100	0.009	0.012	G42	0.003	0.001	0.002	0.992	0.002
G20	0.020	0.893	0.081	0.004	0.003	G43	0.088	0.004	0.064	0.823	0.022
G21	0.278	0.701	0.015	0.003	0.003	G44	0.355	0.006	0.003	0.631	0.005
G22	0.003	0.988	0.003	0.002	0.004	G45	0.246	0.003	0.013	0.735	0.002
G23	0.005	0.984	0.004	0.002	0.004						

**Table 10 genes-13-01147-t010:** Expected heterozygosity (He) and F_ST_ values in four squash subpopulations.

Subpopulation (K)	Expected Heterozygosity (He)	F_ST_
1	0.3210	0.0002
2	0.1858	0.4371
3	0.1947	0.4061
4	0.1567	0.6372
5	0.1907	0.5440
Mean	0.2103	0.4049

## Data Availability

Data are contained within the article.

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
