# Peer review of "iPBS-Retrotransposon Markers in the Analysis of Genetic Diversity among Common Bean (Phaseolus vulgaris L.) Germplasm from Türkiye"

_genes, 2022, doi:10.3390/genes13071147_

Round 1

Reviewer 1 Report

The authors analyzed the genetic diversity of 45 individuals of a critical legume crop (Phaseolus vulgaris L.) in a region of Turkey using iPBS-Retrotransposon Markers. In general, this study was conducted with rational processes. However, the manuscript lacks some details. I think that it could be improved from the revision process. Please see my comments below.

[1] Please consider changing the title since this study did not cover all geographic regions of Turkey.

[2] Line 21, please add the scientific name here.

[3] Line 27-28, the meaning of this sentence is not comprehensible to me. Please revise this sentence.

[4] In this study, the authors often used the term “genotype”. However, its meaning is not clear. How is it different from “individual”?

[5] Introduction, please clarify the rationale for choosing iPBS-Retrotransposon markers for this study.

[6] Lines 101-102, please specify which studies are the “previous studies” here by citing them.

[7] Line 103, please specify what the problem was.

[8] Introduction, please clarify the rationale for choosing the Erzurum-Ispir district for this study.

[9] Lines 112-116, please clarify what the sampling strategy was? How were 45 individuals selected (random?)? Were those seeds from naturalized plants or cultivated plants?

[10] Please provide more geographic information, such as the map of Turkey showing where the Erzurum-Ispir district and collected locations are. GPS information of the collection locations would be valuable too.

[11] Figure 1, please add an explanation of the red line in the figure legend.

[12] Figure 2, please add appropriate explanations in the figure legend.

[13] Figure 3, please add appropriate explanations in the figure legend. What are A, B, C, D, and E here? Subpopulations I- V?

[14] Table 9, Why values for Subpopulation V from G24 to G38 are omitted here?

[15] Please consider adding genotyping dataset as supplementary material.

Author Response

Responses to Comments of Editor

General Response:

Dear editor; According to the valuable suggestions of the commentators, the comments of Reviewer 1 on the manuscript are highlighted in yellow and the comments of Reviewer 2 are highlighted in yellow. Thank you for giving us the chance to review our manuscript. In the table below, we tried to respond to the suggestions and comments of all the referees in the best way possible.

Comments Reviewer 1

Dear reviewer, thank you very much for your valuable suggestions and comments. The manuscript in the article has been corrected again, considering your valuable suggestions. We have also tried to respond in the best way possible to all corrections to your comments and suggestions. The table below contains our responses to suggestions and comments.

Comment

1.       Please consider changing the title since this study did not cover all geographic regions of Turkey.

Response: As per your suggestion was corrected in Title

iPBS-Retrotransposon Markers in the Analysis of Genetic Diversity among Common Bean (Phaseolus vulgaris L.) Germplasm from Northeastern Anatolia region of Turkiye

Comment

2.       Line 21, please add the scientific name here.

Response: As per your suggestion was added scientific name.

Added: “……40 Turkish bean (Phaseolus vulgaris L.)…”

Comment

3.       Line 27-28, the meaning of this sentence is not comprehensible to me. Please revise this sentence.

Response: This line was removed.

Comment

4.       In this study, the authors often used the term “genotype”. However, its meaning is not clear. How is it different from “individual”?

Response: Thank you for your valuable suggestions. According to your suggestions, the term "genotype" was changed to "varietie" throughout the manuscript.

Comment

5.       Introduction, please clarify the rationale for choosing iPBS-Retrotransposon markers for this study.

Response: Dear reviewer, thank you for your suggestion.

-Considering your suggestion, the following sentence was added to the Introduction:

" In our previous studies [26, 29] and in the studies of other researchers [8, 9], it has been observed that retrotransposon markers are quite efficient for genetic diversity studies in terms of the total number of amplified and polymorphic bands."

-In addition, sentences about why iPBS markers are important are available in the introduction:

“Also, among them, retrotransposons are genetic elements capable of forming major components of most eukaryotic genomes, constituting 50-90% of the plant genome. Re-trotransposons are divided into two: long terminal repeat (LTR) and non-LTR retrotrans-posons. LTR-retrotransposons are more common in plants than the other group [24]. Due to limitations in both LTR and non-LTR retrotransposons, inter Primary Binding Site (iPBS) retrotransposons have been developed as a universal marker used in the characterization of both animal and plant species [27]. iPBS markers are the dominant markers and have become a preferred marker in genetic diversity assessment in recent years due to their universality [25]. The universality of the iPBS-retrotransposon marker has been proven and molecular characterization and phylogenetic studies are available for these markers, also in beans [8, 24, 26].”

Comment

6.       Lines 101-102, please specify which studies are the “previous studies” here by citing them.

Response: As per your suggestion was added reference in sentence.

Reference added; “Previous studies [7, 8, 9, 20, 26]…….

Comment

7.       Line 103, please specify what the problem was.

Response: As per your suggestion was corrected

“Therefore, these studies were carried out to determine the genetic diversity and population structure of bean in İspir district of Turkey, where the study was conducted.”

Comment

8.       Introduction, please clarify the rationale for choosing the Erzurum-Ispir district for this study.

Response: A new sentence has been added according to your suggestions.

“In Erzurum-İspir district, there are no previous studies to reveal bean genetic diversity and population structure. These studies were carried out to determine the genetic diversity and population structure of bean in İspir district of Turkey, where the study was conducted. Therefore, we here investigated the genetic diversity and population structure local bean varieties collected from the district of Ispir, using the iPBS marker system.”

Comment

9.       Lines 112-116, please clarify what the sampling strategy was? How were 45 individuals selected (random?)? Were those seeds from naturalized plants or cultivated plants?

Response: Considering your valuable suggestions, necessary explanations have been added to the Material section.

“In this study, 45 Turkish bean (Phaseolus vulgaris L.) local varieties were used as plant material. The names and gathering places of the regional varieties are presented in Table 1 and Figure 1. Bean local varieties were collected in cultivated fields in 8 different İspir districts of Erzurum in the northeastern Anatolia region of Turkey.”

Comment

10.    Please provide more geographic information, such as the map of Turkey showing where the Erzurum-Ispir district and collected locations are. GPS information of the collection locations would be valuable too.

Response: Considering your valuable suggestions, a map showing the geographical location has been added (Figure 1). The numbers of the other shapes have been updated as the new shape has been added. In addition, GPS information has been added to Table 1.

Comment

11.    Figure 1, please add an explanation of the red line in the figure legend.

Response: Considering your valuable suggestions, the figure has been updated and the red line has been removed.

Comment

12.    Figure 2, please add appropriate explanations in the figure legend.

Response: Figure description has been edited considering your valuable suggestions.

Figure 3. Principal coordinates analysis (PCoA) calculated from the pooled data of twenty-six inter-Primer Binding Site (iPBS) primers in 45 bean varieties.

Comment

13.    Figure 3, please add appropriate explanations in the figure legend. What are A, B, C, D, and E here? Subpopulations I- V?

Response: Added roman numerals denoting subpopulations instead of letters. (I-V)

Comment

14.    Table 9, Why values for Subpopulation V from G24 to G38 are omitted here?

Response: Thank you for your comment. V. population information of the 24-38 varieties has been added to the table.

Comment

15.    Please consider adding genotyping dataset as supplementary material.

Response: The information of the genotyping dataset was added to the system as supplementary material.

Comments Reviewer 2

Dear reviewer, thank you very much for your valuable suggestions and comments. The manuscript in the article has been corrected again, considering your valuable suggestions. We have also tried to respond in the best way possible to all corrections to your comments and suggestions. The table below contains our responses to suggestions and comments.

Comment

1.       Descriptive study of iPBS generated polymophism among bean accessions. Generally well written. Based on the retrotransposon characteristics, possible effect of their activation should be at least discussed - please, add this and summarize in the results, what effect in the grouping of the analysed genotypes it most relevant. Conclusions shoul be more adressed.

Response: Dear reviewer; Thank you very much for your valuable suggestions. We have tried to improve the manuscript as much as we can, taking into account your comments and suggestions. Your edits are highlighted in yellow on the manuscript.

Sincerely

Reviewer 2 Report

Descriptive study of iPBS generated polymophism among bean accessions. Generally well written. Based on the retrotransposon characteristics, possible effect of their activation should be at least discussed - please, add this and summarize in the results, what effect in the grouping of the analysed genotypes it most relevant. Conclusions shoul be more adressed.

Author Response

(The authors gave the same response as above.)
